# E.PathDash, pathway activation analysis of publicly available pathogen gene expression data

Lily Taub,[1] Thomas H. Hampton,[1] Sharanya Sarkar,[1] Georgia Doing,[2] Samuel L. Neff,[3] Carson E. Finger,[1] Kiyoshi Ferreira Fukutani,[1] Bruce A. Stanton[1]

**ABSTRACT** E.PathDash facilitates re-analysis of gene expression data from pathogens clinically relevant to chronic respiratory diseases, including a total of 48 studies, 548 samples, and 404 unique treatment comparisons. The application enables users to assess broad biological stress responses at the KEGG pathway or gene ontology level and also provides data for individual genes. E.PathDash reduces the time required to gain access to data from multiple hours per data set to seconds. Users can download high-quality images such as volcano plots and boxplots, differential gene expression results, and raw count data, making it fully interoperable with other tools. Importantly, users can rapidly toggle between experimental comparisons and different studies of the same phenomenon, enabling them to judge the extent to which observed responses are reproducible. As a proof of principle, we invited two cystic fibrosis scientists to use the application to explore scientific questions relevant to their specific research areas. Reassuringly, pathway activation analysis recapitulated results reported in original publications, but it also yielded new insights into pathogen responses to changes in their environments, validating the utility of the application. All software and data are freely accessible, and the application is available at scangeo.dartmouth.edu/EPathDash.

**IMPORTANCE** Chronic respiratory illnesses impose a high disease burden on our communities and people with respiratory diseases are susceptible to robust bacterial infections from pathogens, including *Pseudomonas aeruginosa* and *Staphylococcus aureus*, that contribute to morbidity and mortality. Public gene expression datasets generated from these and other pathogens are abundantly available and an important resource for synthesizing existing pathogenic research, leading to interventions that improve patient outcomes. However, it can take many hours or weeks to render publicly available datasets usable; significant time and skills are needed to clean, standardize, and apply reproducible and robust bioinformatic pipelines to the data. Through collaboration with two microbiologists, we have shown that E.PathDash addresses this problem, enabling them to elucidate pathogen responses to a variety of over 400 experimental conditions and generate mechanistic hypotheses for cell-level behavior in response to disease-relevant exposures, all in a fraction of the time.

**KEYWORDS** bioinformatics, gene expression, respiratory pathogens, pathway analysis

In a review of the 2017 Global Burden of Disease report conducted by the Institute for Health Metrics and Evaluation, Soriano et al. found that the prevalence of chronic respiratory diseases worldwide increased by 39.8% (1), and recent research has identified an association between multiple chronic respiratory diseases and changes in the airway microbiome (2). Therefore, in developing treatments for chronic respiratory diseases, it is important to understand how pathogens respond to experimental conditions such as growth medium, drug treatment, or genetic mutation (3–5). Although there have

Address correspondence to Lily Taub, lily.d.taub@dartmouth.edu.

The authors declare no conflict of interest.

See the funding table on p. 19.

been thousands of studies in this area of research, only a small percentage of the archived data is readily accessible for re-analysis. The goal of the application described here is to facilitate both gene and pathway-level re-analysis of publicly available gene expression data from pathogens clinically relevant to a variety of respiratory diseases, including *Pseudomonas aeruginosa*, *Bacteroides thetaiotaomicron*, *Staphylococcus aureus*, and *Streptococcus sanguinis* (6–15).

The rise in transcriptomic data production began with the invention of the microarray in 1995 (16) and has continued through the past three decades, with data accumulating ever more rapidly after the development of sequencing methodologies that utilize the computational and hardware advances of modern computing (17). Public repositories, such as the European Bioinformatics Institute's ArrayExpress (18) and the National Center for Biotechnology Information's Gene Expression Omnibus (GEO) (19), have come online in concert with the rise of high-throughput sequencing data. As the infrastructure for data collection and hosting has become more robust, the data science and research communities have adopted the mission of developing findable, accessible, interoperable, reusable (FAIR) data practices (20). While public repositories further the goals of findability and accessibility, the microbiology research community would nonetheless benefit from easy to use data reuse platforms that do not require statistical expertise or computationally expensive and labor-intensive data cleaning and formatting.

The goal of the application described in this paper is to facilitate the reuse of publicly available data. Our application reduces the time required to gain access to data in 48 studies, 548 samples, and 404 unique treatment comparisons from multiple hours per data set to seconds. The application primarily serves the research community of microbiologists interested in pathogens associated with chronic respiratory diseases. In this paper, we chose cystic fibrosis (CF) as a case study to demonstrate the value of the application to researchers. People with CF (pwCF) are subject to chronic lung infections, which are responsible for 90% of the disease's morbidity and mortality (21–24). Two pathogens that commonly dominate the lungs of pwCF, *Pseudomonas aeruginosa* (*P. aeruginosa*) and *Staphylococcus aureus* (*S. aureus*), are included in over 80% of the data sets in the application presented here. The 2022 CF Patient Registry reported *S. aureus* infection in 60%–80% of CF patients in age cohorts younger than 18 years old (25). Research has shown a strong association between *S. aureus* infection and poor clinical outcomes like decreased lung function and increased inflammation (26, 27). *P. aeruginosa,* dominant in late-stage lung infections, has been shown to be multi-drug resistant and is adept at forming biofilms that inhibit the immune response to bacterial infection (28).

Both *S. aureus* and *P. aeruginosa* are also clinically relevant beyond CF. As members of the ESKAPE pathogen group (*Enterococcus faecium*, *Staphylococcus aureus*, *Klebsiella pneumoniae*, *Acinetobacter baumannii*, *Pseudomonas aeruginosa*, *Enterobacter* sp.), they are classified as highly virulent organisms that are adept at developing antibiotic resistance (29, 30). There is a demonstrated association between *S. aureus* and persistent asthma (14) and methicillin-resistant *S. aureus* (MRSA) is a common cause of hospital-acquired pneumonia infections (31). In chronic obstructive pulmonary disease (COPD), *P. aeruginosa* infection has been shown to increase the risk of exacerbation and hospitalization events (32) (Table S3).

E.PathDash (ESKAPE Act PLUS Pathway Dashboard) streamlines the re-analysis of publicly available RNA-Seq data relevant to respiratory diseases by enabling users to run pathway activation analysis for all possible treatment comparisons within each data set in its compendium. This analytical approach uses transcriptomic data to identify differentially activated or repressed biological pathways across treatment comparisons. In doing this, the program contributes to the body of tools that promote FAIR data principles and allows users to derive biological insights and formulate new hypotheses for future experiments. It achieves this without requiring investigators to be familiar with statistical methods for pathway analysis, or forcing them to normalize raw gene counts, establish consistent gene identifier encodings, or perform differential gene expression

analysis. E.PathDash dramatically reduces the time required to access and analyze the data included in its compendium from many hours per data set to seconds.

E.PathDash can be accessed freely at scangeo.dartmouth.edu/EPathDash.

## Addressing the limitations of existing applications

A number of bioinformatic tools that perform pathway analysis have been developed in response to the increased use of omics data in biological research (33). Most of the applications require user-supplied data and, therefore, do not readily promote the reuse of publicly available datasets (34–37). An exception to note is iDEP (integrated Differential Expression and Pathway analysis), but the data sets included in iDEP do not include data from bacterial species of interest to respiratory disease researchers (37).

We have previously published several applications that facilitate computational tasks for microbiologists and CF researchers (36, 38–40). Two of these applications, ESKAPE Act PLUS (Activation Analysis for ESKAPE Pathogens and other Prokaryotes Labs Usually Study) (36) and CF-Seq (38), inspired the development of the application described here. ESKAPE Act PLUS performs pathway activation analysis on user-uploaded differential gene expression data. It uses the same binomial test method employed in E.PathDash, which is described fully in Materials and Methods. CF-Seq is a platform for exploring gene-level analyses of public experimental data from clinically relevant CF pathogens, utilizing RNA-Seq data sets from the GEO. By combining the statistical backend of ESKAPE Act PLUS and a subset of relevant processed RNA-Seq data from CF-Seq, E.PathDash gives users a platform to explore pathway activation across a curated compendium of publicly available RNA-Seq data sets without having to perform any data cleaning, formatting, or analysis tasks themselves. This saves users a considerable amount of time even if they have the skills to perform the steps required to access the data.

## RESULTS

E.PathDash is a R Shiny web application that facilitates the re-analysis of publicly available pathogen gene expression data relevant to respiratory diseases. Its compendium includes a total of 48 studies, 548 samples, and 404 unique treatment comparisons. Specifically, the application automates pathway activation analysis for each RNA-Seq data set in its compendium, all of which were originally sourced from the GEO. Figure 1 provides an overview of the application flow, capturing the decisions users make as they navigate the application (orange branch nodes) and the analysis products that are generated (purple leaf nodes).

### Workflow

After launching the application in an internet browser and reading an application overview (Fig. 2, screen 1), the user begins interacting with the compendium data by filtering the cataloged data sets on bacterial species and strain(s) of interest (Fig. 2, screen 2). From here, the user can navigate between four different dashboard pages (Fig. 2, screens 3–5). Individual screenshots of the dashboard pages are included in the supplemental material (Fig. S1 to S6).

### Study Explorer

Selecting a single data set (labeled by GEO accession number) from a list filtered by bacterial species and strain (Fig. 2, label C) activates the Study Explorer page (Fig. 2, screen 3). This page contains metadata about the data set (a link to the GEO entry, study title, study description), downloadable raw counts and study design matrices, and four pathway activation analysis components controlled by a treatment comparison drop down (Fig. 2, screen 3). The treatment comparison considers all samples exposed to the two specified experimental conditions, and the analyses identify systematic transcriptional changes between these sample groups. Significantly activated or repressed Kyoto

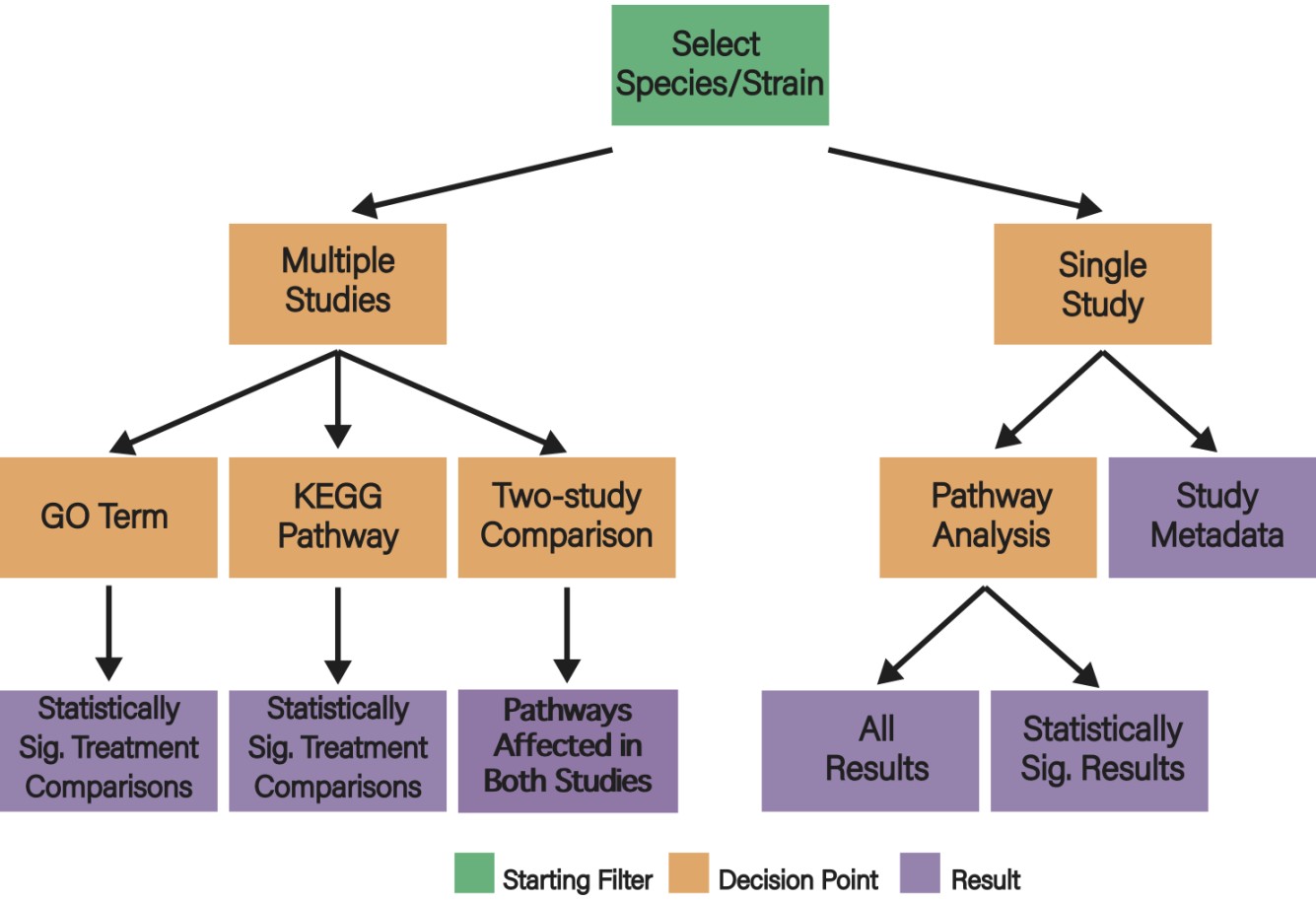

**FIG 1** Conceptual flowchart for E.PathDash. The root (green) represents the bacterial species/strain filter the user must set to start exploring the data. Orange nodes represent decision points the user makes while navigating the application, and the purple leaves are results the different decision paths generate.

Encyclopedia of Genes and Genomes (KEGG) pathways (41, 42) and gene ontology (GO) terms (43, 44) are shown in two boxplots, which use the distribution of $\log_2$ fold change (logFC) values for genes within the pathway to capture how the pathway is activated or repressed (Fig. 2, label E). Two tables, one for KEGG pathways and one for GO terms, contain additional statistical information regarding the pathway activation analysis: binomial test statistic (which represents the proportion of activated pathway genes, i.e., those with a positive logFC value), median gene logFC, *P*-value, and FDR corrected *P*-value (Fig. 2, label F). Changing the treatment comparison (Fig. 2, label D) updates the data in the plots and tables to show pathway activation analysis for the specified comparison. For example in treatment 1 vs treatment 2, treatment 2 is the reference and the results can be interpreted as follows: pathways are activated or repressed in treatment 1 compared to treatment 2.

**KEGG Pathway Explorer and GO Term Explorer**

Users can search the data sets based on a KEGG pathway of interest in the KEGG Pathway Explorer page, which provides an interface to compare activation between data sets. The bacterial species and strain filters control the pathway options available. The pathways have to be defined by KEGG for the given organism and be significantly activated or repressed in at least one treatment comparison. Selecting a pathway returns a table that lists all data sets for which that pathway was significantly activated or repressed (Fig. 2, label G). In addition to the GEO accession number, the table contains the specific treatment comparison that showed activation or repression and the median logFC value for the genes within the pathway. Selecting a row in the results table renders a volcano

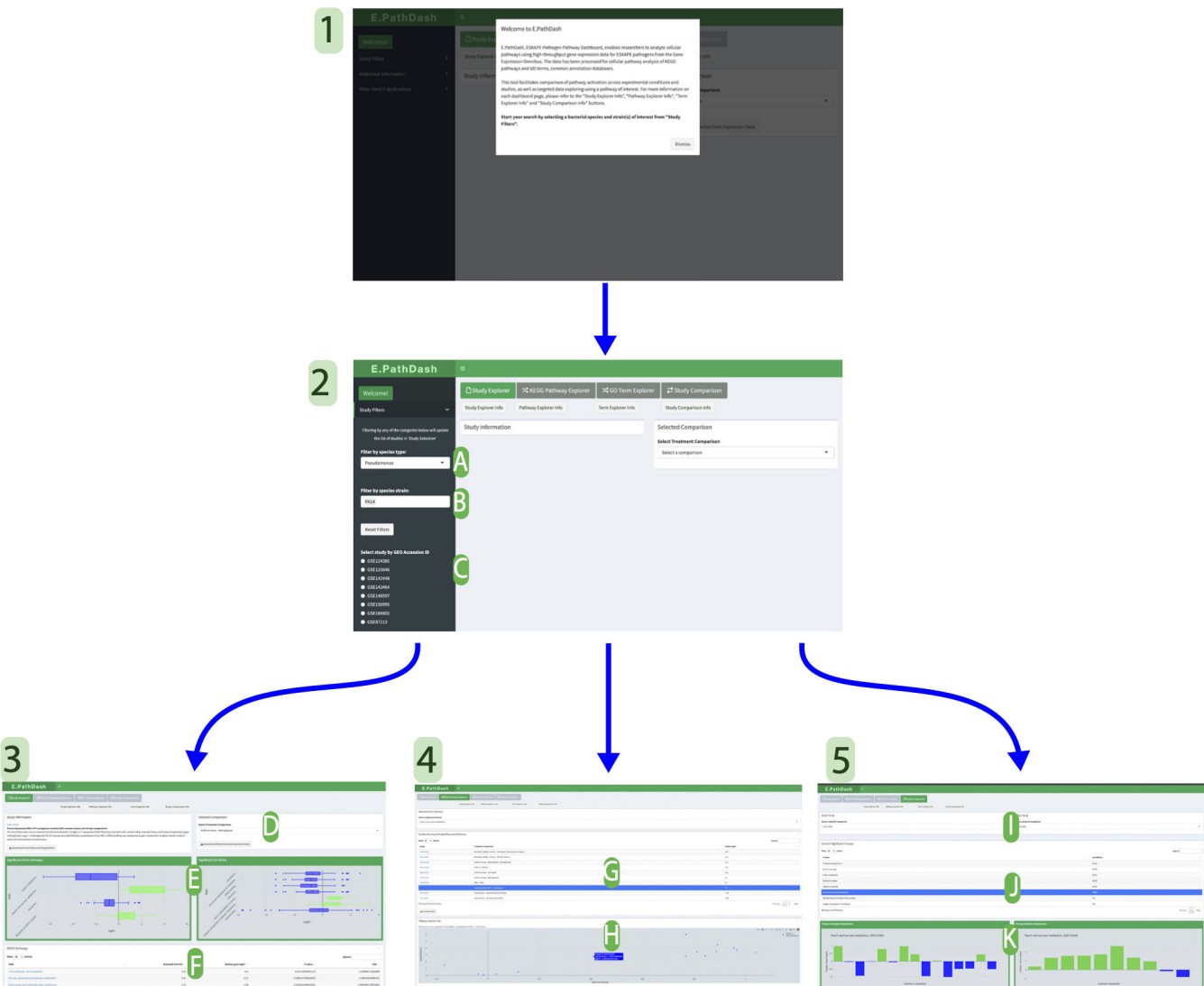

**FIG 2** Screen 1: landing page. Screen 2: filtering side panel where users select species (A) and strain (B). Studies that meet filter requirements are shown in panel C. After filtering users can move to screen 3, 4, or 5 by selecting the corresponding button. Screen 3: the Study Explorer page shows information about a selected study. Selecting a treatment comparison (D) populates boxplots of gene expression in statistically significant activated or repressed KEGG pathways and GO terms (E) and a table of statistical information for all pathways/terms analyzed (F). Screen 4: the KEGG Pathway Explorer and GO Term Explorer pages search the data by a pathway or term of interest and return a table of studies in which the pathway was significantly expressed (G). Selecting a study shows a volcano plot for the genes within the pathway (H). Screen 5: in the Study Comparison page, users select two studies (I) to see commonly activated/repressed KEGG pathways and GO terms (J). Selecting a pathway renders bar charts of median gene logFC value for each treatment comparison (K). There are enlarged versions of the dashboard pages in the supplemental material (Fig. S1-S6).

plot of the genes within the pathway of interest. The volcano plot compares a gene's logFC value on the *x*-axis to the negative $\log_{10}$ transformed *P*-value on the *y*-axis. The *P*-value transformation plots statistically significant genes at higher *y*-axis values. This feature can be used in tandem with the Study Explorer page to investigate how genes within an activated or repressed pathway were differentially expressed in the study and treatment comparison of interest. This functionality is duplicated in the GO Term Explorer page, where users can search the data by GO term.

## Study Comparison

The Study Comparison page allows users to compare pathway activation analyses for two selected data sets (Fig. 2, screen 5). The page contains a table of KEGG pathways and

GO terms that were significantly activated or repressed in both study data sets (Fig. 2, label J). Selecting an entry in the table shows how the KEGG pathway or GO term was expressed across the different treatment comparisons within each data set (Fig. 2, label K), utilizing the bar plot described in "Application design" in Materials and Methods.

## Downloadable content

Downloadable content is an important feature of the application because it provides full interoperability with other tools. Users can download all generated graphs, tables, raw gene count data, and differential gene expression results for each treatment comparison within each data set. Furthermore, users can download the differential gene expression data (by study and sample comparison) for just the genes within a KEGG pathway or GO term of interest. A complete list of downloadable content is included in the supplementary material (Table S1). Users can leverage this clean and well-formatted data to jump start additional analyses they wish to perform in other bioinformatic pipelines, eliminating the significant pre-processing time necessary to make much publicly available transcriptomic data usable.

E.PathDash is able to recapitulate findings from studies associated with the data sets in its compendium. The study by Farrant et al. (GEO data set GSE124385) reported on activation of the sulfur metabolism pathway and type III secretion system by hypochlorous and hypothiocyanous acids in *P. aeurginosa* (45)*, which was observed in the analyses from E.PathDash (Table S4; Fig. S7). Additionally, Bouzo et al. (GEO data set GSE142448) reported that manuka honey treatment depressed quorum sensing and fatty acid metabolism in *P. aeruginosa* (46), again findings that were seen in E.PathDash pathway analyses (Table S4). To demonstrate the capabilities of the application beyond confirming findings from the literature, we gave it to two CF scientists, who used the application to investigate questions relevant to their research. They contributed user stories detailing how they used the application, their findings, and their overall experience with E.PathDash.

## Case study #1: propanoate metabolism in *P. aeruginosa* during microbial interactions (Dr. Georgia Doing, The Jackson Laboratory for Genomic Medicine)

To expand upon conclusions drawn from a previously published data set of *P. aeruginosa* gene expression from *in vitro* co-culture with *Candida albicans* (GEO: GSE148597) (47), I used E.PathDash to explore a comparison between *P. aeruginosa* in co-culture and mono-culture (in the application, the treatment comparison selected was "co-culture WT - PA Monoculture"). This comparison is complementary to those presented in the original publication and reveals exciting new hypotheses. Re-analyzing the data using the E.PathDash Study Explorer shows that many pathways are activated in *P. aeruginosa* when grown on *C. albicans* including many pathways involved in metabolizing amino acids, sugars, and carboxylic acids. The availability of amino acids and sugars may differ across mono-culture and co-culture conditions, and the activation of these pathways is likely due to competition for the resources of the underlying medium, whereas metabolic processes involving carboxylic acids are more likely reflective of microbial interactions via the exchange of metabolic intermediates and exoproducts. Furthermore, given the broad activation of many pathways, these results suggested there may be simultaneous or interacting pathways not previously implicated in *P. aeruginosa*'s response to *C. albicans*. Due to the known role of ethanol as an exchanged metabolite, I checked for pathways that potentially interact with ethanol metabolism via shared genes but also suggest the exchange of novel metabolites. In fact, the E.PathDash re-analysis revealed the propanoate metabolism KEGG pathway (pau00640) had statistically significant activation (binomial test estimate 0.8, median gene fold change 0.99, and FDR < 0.001). Specifically, *P. aeruginosa* genes for the metabolism of propanoate have higher mRNA levels when *P. aeruginosa* is in co-culture with *C. albicans* compared to when *P. aeruginosa*

is in mono-culture (Fig. 3A). While propanoate metabolism may have been previously overlooked due to key enzymes being attributed solely to ethanol catabolism, this exploratory re-analysis suggests *P. aeruginosa* may also be metabolizing propanoate or inducing the 2-methylcitrate cycle (48).

Changing the treatment comparison in the Study Explorer showed that the KEGG propanoate metabolism pathway is also activated, though to a lesser extent, when *P. aeruginosa* is grown in co-culture with WT *C. albicans* compared to *P. aeruginosa* grown with the *adh1Δ/Δ C. albicans* mutant, which is deficient in ethanol production (Fig. 3B) (binomial test estimate 0.88, median log fold change 0.53, and FDR < 0.0001). The trend toward increased expression of propanoate metabolism genes when *P. aeruginosa* is grown with WT *C. albicans* compared to *adh1Δ/Δ C. albicans*, as well as the lower magnitude fold change when *P. aeruginosa* is grown with *adh1Δ/Δ C. albicans* compared to *P. aeruginosa* grown in mono-culture (binomial test estimate 0.63, median log fold change 0.53, and FDR 0.24), suggests that *C. albicans* ethanol production may be contributing to the response of increased propanoate metabolism in *P. aeruginosa*. This is consistent with a report in *E. coli* that showed ethanol-induced propanoate metabolism activated the expression of *prpD*, a gene also present in *P. aeruginosa* (49).

Downloading the log fold change values of the genes in the propanoate metabolism KEGG pathway using the E.PathDash KEGG Pathway Explorer interface (Table S2) provided the nuanced information on the top five genes in the propanoate KEGG

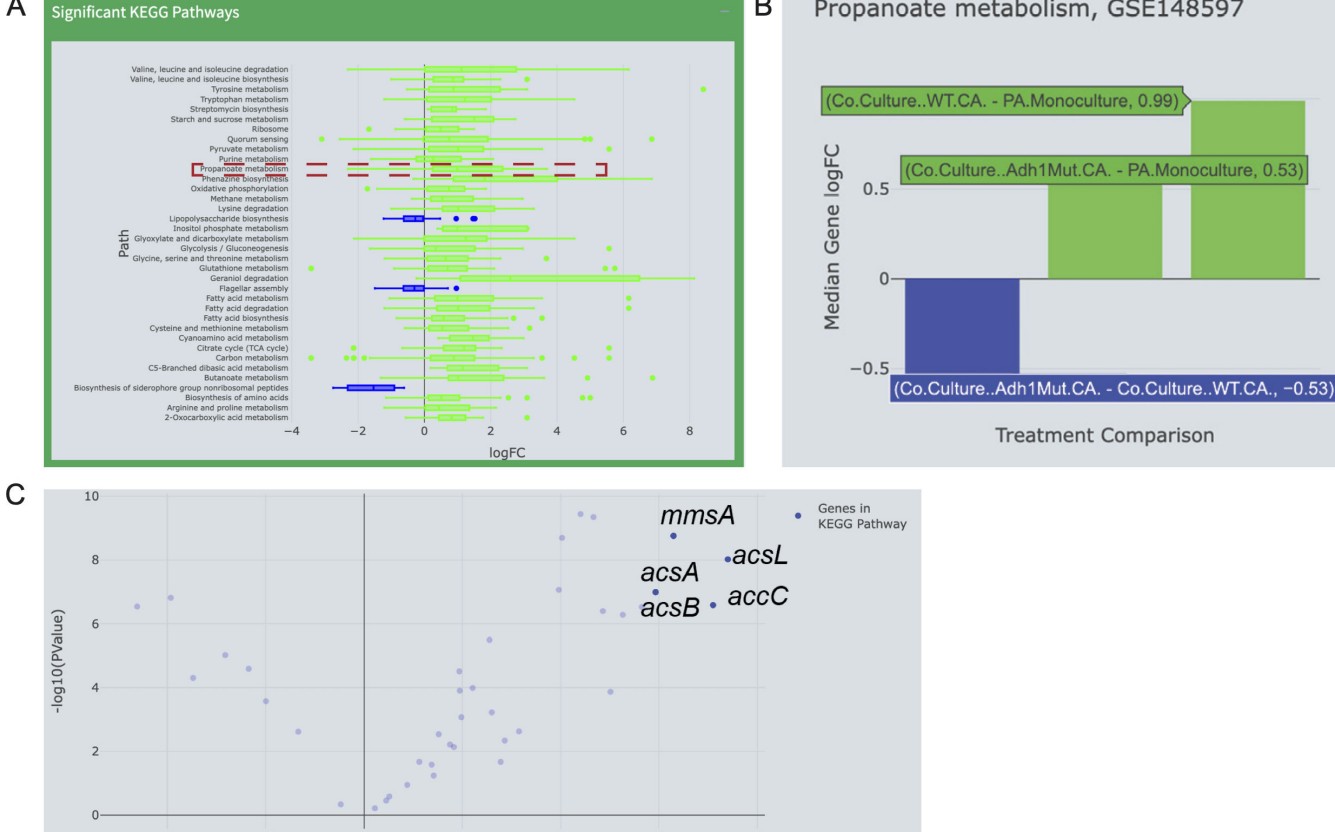

**FIG 3** Use of E.PathDash to analyze the expression of propanoate metabolism genes in *P. aeruginosa*. (A) The Study Explorer of GSE148597 shows that the KEGG pathways significantly activated in *P. aeruginosa* co-cultured with *C. albicans* compared to mono-culture include propanoate metabolism. Box and whisker plots show log fold change distributions of genes in each pathway. (B) The Study Comparison tool shows the other comparisons in the study and suggests *C. albicans* ADH1-dependent ethanol production plays a role in stimulating the expression of the propanoate metabolism KEGG pathway. (C) Volcano plot from the Study Explorer shows individual genes in the propanoate metabolism KEGG pathway. Data are plotted as fold change values and negative log transformed *P*-values.

pathway. The highest log fold change values in co-culture were *acsL*, *accC*, *mmsA*, *acsA*, and *acsB* (Fig. 3C). Notably, *acsL*, *acsA,* and *acsB* are acetyl-CoA synthetases that can convert propanoate to propionyl-adenylate and then propanoyl-CoA, but *acsA* and *acsB* can also convert acetate to acetyl-CoA during ethanol oxidation (50). This suggests that ethanol may induce ErdR responsive genes (51) which, in addition to regulation through *prpD*, may also facilitate the metabolism of propanoate.

The E.PathDash KEGG Pathway Explorer shows that propanoate metabolism is also activated in a study, wherein *P. aeruginosa* was treated with farnesol (GEO: GSE138731, median gene log fold change 0.26, Fig. 4A), a *C. albicans*-produced quorum-sending molecule known to alter *P. aeruginosa* metabolism (52). Furthermore, propanoate metabolism is also activated in studies in which *P. aeruginosa* is transitioned from anaerobic conditions to microaerobic conditions (GEO: GSE71880, median gene log fold change 0.63, Fig. 4B) and treated with artificial honey and methylglyoxal (GEO: GSE142448, median gene log fold change −1.22, Fig. 4C) (46). The E.PathDash Study Comparison feature made the specific directionality of each comparison clear: propanoate was activated upon the addition of farnesol and micro-oxia and repressed upon the addition of methylglyoxal (Fig. 4, bars highlighted in red). These conditions may mimic those of metabolic imbalance such that intermediates of glycolysis or the TCA cycle accumulate. Notably, methylglyoxal can be detoxified into intermediates that converge with those of propanoate metabolism (53), and the TCA intermediate succinate can stimulate propanoate metabolic genes (54). While these connections are likely indirect, they provide preliminary evidence suggesting co-culture with *C. albicans* alters *P. aeruginosa* central metabolism in a manner that leads to altered pools of metabolic intermediates and perhaps a convergence on propanoate metabolism.

Analysis of previously published data sets with E.PathDash suggests a hypothesis, wherein *P. aeruginosa* propanoate metabolism may be stimulated by *C. albicans*-produced ethanol through (i) direct regulation of propanoate catabolic gene *prpD* by ethanol-sensing ErdR and (ii) metabolic convergence of ethanol oxidation and propanoate catabolism by dual-function acetyl-CoA synthetases *acsA* and *acsB* (Fig. 5). This model could be readily tested with *erdR* mutations and assays of *acsA* and *acsB* expression as well as growth on ethanol and propanoate. Additionally, the broad activation of the propanoate KEGG pathway in other studies that looked at *in vitro* conditions relevant to co-culture suggest propanoate and ethanol metabolism may be further stimulated by (iii) other factors present in co-culture such as farnesol, oxygen limitation, and metabolic intermediate methylglyoxal (Fig. 5). The pathway-level interactions could be untangled with epistasis experiments on relevant environment-sensing genes such as oxygen-sensing *anr* and metabolic genes like methylglyoxal reductases. This tool facilitated hypothesis generation resulting in a testable model of the key role propanoate metabolism may play in medically relevant microbial interactions of *P. aeruginosa*.

## Case study #2: differential effects of DNA-gyrase inhibitor classes on biofilm formation in *P. aeruginosa* (Sharanya Sarkar, Ph.D. candidate, Geisel School of Medicine at Dartmouth)

*P. aeruginosa* is the most common pathogen found in adult people with cystic fibrosis (pwCF) (25). To maintain control of infections, pwCF are prescribed antibiotics as a part of their treatment regimen (55) and DNA gyrase inhibitors, particularly the fluoroquinolones, constitute a standard treatment for *P. aeruginosa* infections (56). One of the most challenging problems with *P. aeruginosa* infections is the formation of robust, antibiotic-resistant biofilms within the respiratory tract that are exceedingly challenging to eliminate once established (57, 58). Our lab investigates antimicrobial treatments that interfere with biofilm formation in *P. aeruginosa* (59–61). Since gyrase inhibitors also inhibit biofilm formation (62, 63), we used E.PathDash to analyze how DNA gyrase inhibitors affected this process in experimental settings. Additionally, we wanted to

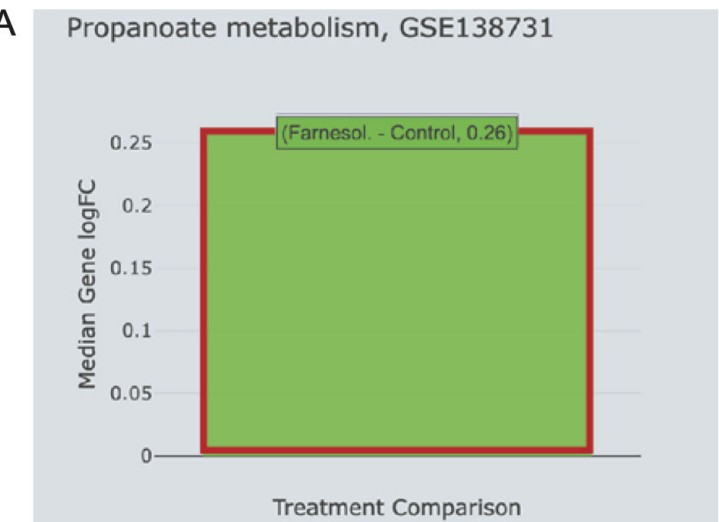

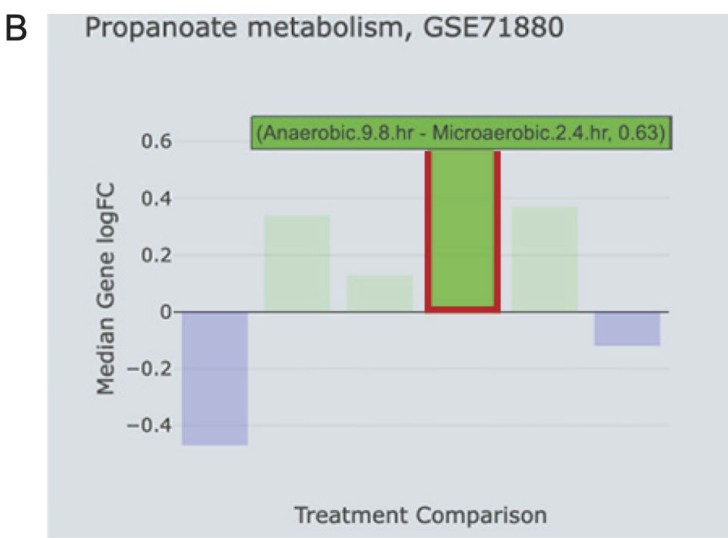

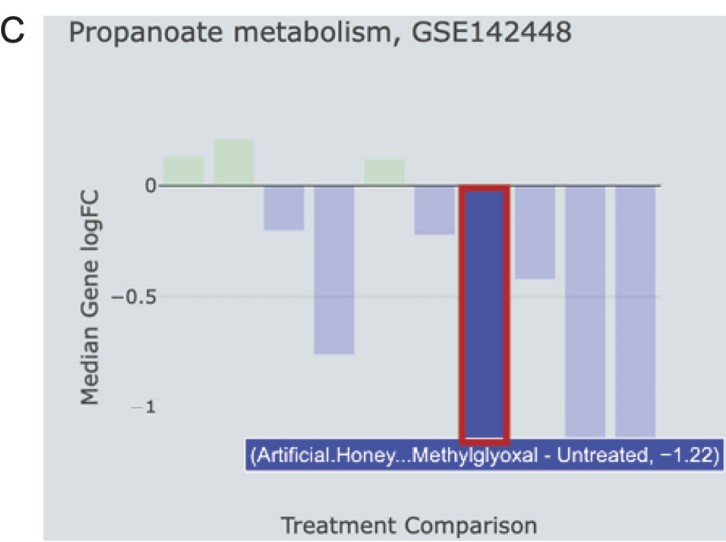

**FIG 4**  The KEGG Pathway explorer showed that propanoate metabolism genes are also increased in expression in *in vitro* studies of (A) *P. aeruginosa* treated with farnesol, (B) transitioned between anaerobic and micro-oxic conditions, and (C) treated with artificial honey and methylglyoxal. Highlighted bars show relevant comparisons of each study.

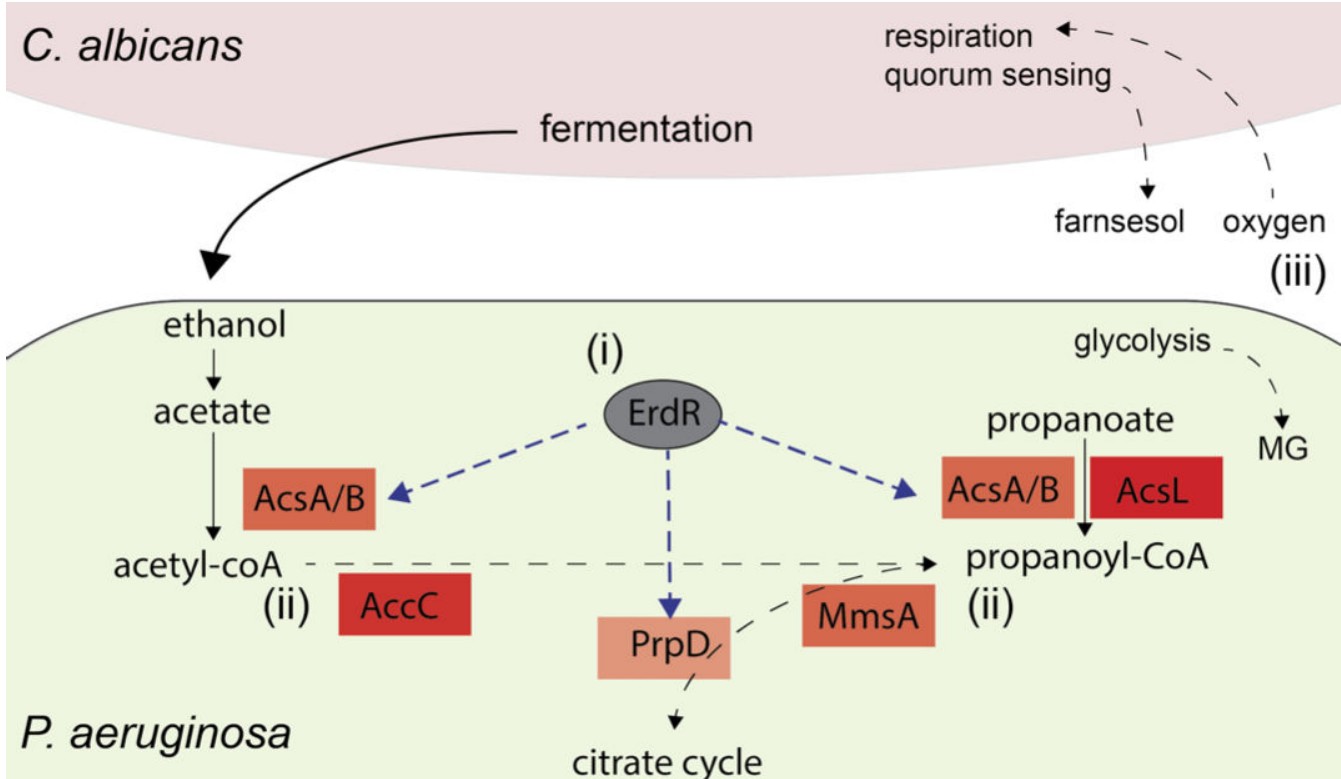

**FIG 5** Hypothesized model synthesizing the results found using E.PathDash: ethanol may stimulate propanoate metabolism directly through ErdR regulation of *prpD* and indirectly through activity of *acsA* and *acsB* acting upon both acetate and propanoate. Furthermore, metabolic products of ethanol and propanoate catabolism may converge on propanoyl-CoA and feed into the citrate cycle.

determine if distinct classes of DNA-gyrase inhibitors impacted biofilm formation differentially.

Utilizing the species filter in E.PathDash to search across study data for *P. aeruginosa* data sets, we investigated data set GSE166602, which looked at molecular signatures in *P. aeruginosa* with different gyrase inhibitors (64). Using different treatment comparisons in the Study Explorer page, we found that untreated *P. aeruginosa* had significantly upregulated biofilm formation compared to *P. aeruginosa* treated with coumermycin, with a median gene logFC value of 0.33 (Fig. 6A, FDR 0.015). In other words, the median expression of biofilm genes in the untreated condition was 1.26 times greater than that observed in the coumermycin condition. Subsequently, we aimed to compare the inhibitory effects of ciprofloxacin, an antibiotic belonging to the fluoroquinolone group (the second class of gyrase inhibitors), with a control condition (untreated). The results showed that ciprofloxacin repressed biofilm formation in *P. aeruginosa* compared to untreated *P. aeruginosa*, with a median gene logFC of −0.22 (Fig. 6B, FDR 0.042). In this comparison, gene expression within the biofilm formation pathway in the untreated condition was 1.16 times greater than that with ciprofloxacin treatment.

Considering these two comparisons, we hypothesized that *P. aeruginosa* treated with ciprofloxacin will have greater biofilm formation compared to those treated with coumermycin. As anticipated, when comparing ciprofloxacin-treated *P. aeruginosa* to coumermycin-treated *P. aeruginosa*, biofilm formation was upregulated in the former group (Fig. 6C, FDR 0.001). This indicates that ciprofloxacin does not substantially reduce biofilm formation, unlike coumermycin.

Next, we explored responses of genes in the biofilm formation KEGG pathway using the KEGG Pathway Explorer page. Our specific focus was on examining significantly upregulated biofilm-related genes in control *P. aeruginosa* exhibiting a minimum log2 fold change of 0.5, as compared to coumermycin. One particular gene (UniProt:

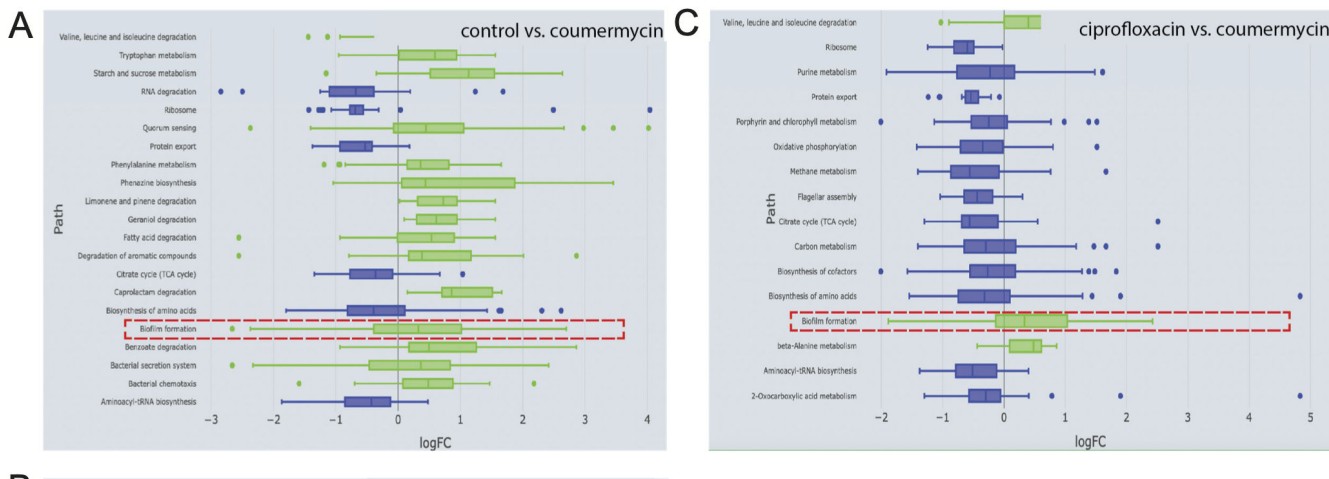

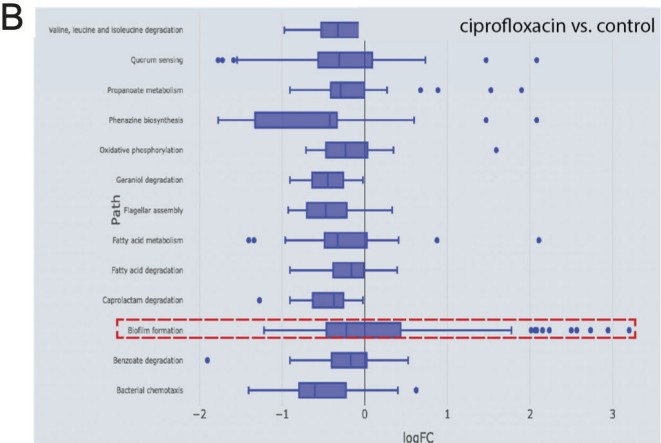

**FIG 6** (A) Coumermycin downregulates biofilm formation in *P. aeruginosa* compared to control, with median expression of biofilm genes 0.79 times that of control (median logFC −0.33). (B) Ciprofloxacin represses biofilm formation in *P. aeruginosa* as compared to control with median expression of biofilm genes 0.85 times that of control (median logFC −0.22). (C) Coumermycin is more efficient than ciprofloxacin in reducing biofilm formation in *P. aeruginosa*.

A0A0H2ZEE6, gene name: *rhlI*) met these criteria (Fig. 7A) and was chosen for further scrutiny. A brief search using the UniProt ID revealed that this gene encodes an Acyl-homoserine-lactone synthase. Previous research has demonstrated that biofilm formation in *P. aeruginosa* can be influenced by targeting acyl-homoserine-lactones (AHLs) (65). Taken together, the analysis suggests that one of the mechanisms through which coumermycin significantly downregulates biofilm formation in

*P. aeruginosa* is by targeting AHL synthase. Given that coumermycin demonstrated superior efficacy in preventing biofilm formation compared to ciprofloxacin, we postulate that the central biofilm gene, AHL synthase, might be more highly expressed in the ciprofloxacin group than in the coumermycin group. We referred to the downloaded gene expression data for the ciprofloxacin-coumermycin comparison from the KEGG Pathway Explorer page and easily located the log2 fold change value for this gene. It was evident that this gene was significantly upregulated in the ciprofloxacin group (Fig. 7B), implying that ciprofloxacin may not be as proficient as coumermycin in targeting AHL synthase.

In summary, the application provided a highly visual and efficient means to re-evaluate a public data set. It indicated that coumermycin could potentially be a more effective option for treating biofilm-forming microorganisms such as *P. aeruginosa*, in comparison to ciprofloxacin, and this can be tested experimentally. Our re-analysis of the data extends the findings of the original publication by focusing on quorum-sensing mediators.

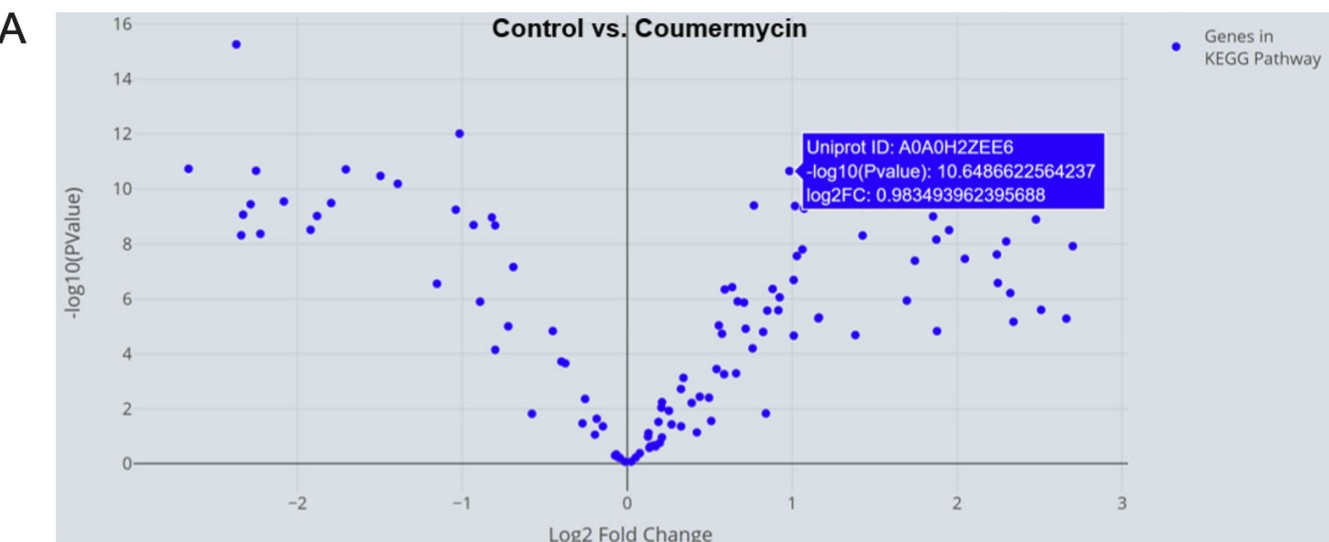

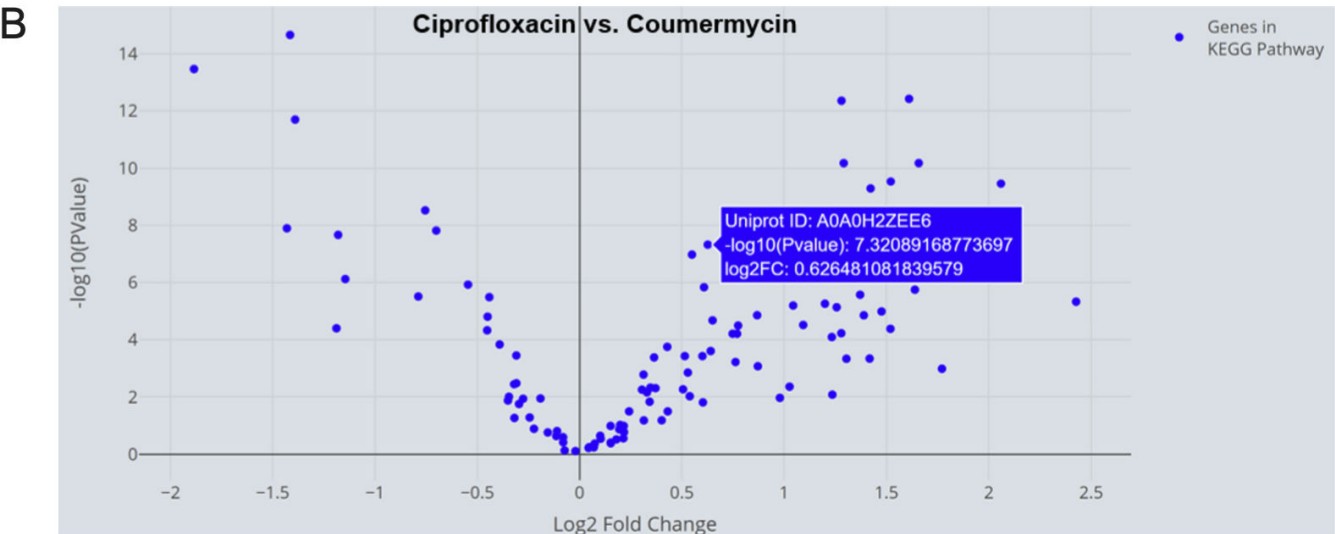

**FIG 7** AHL synthase, a key biofilm gene in *P. aeruginosa*, is significantly upregulated in (A) control, as compared to coumermycin-treated *P. aeruginosa*, and in (B) ciprofloxacin-treated *P. aeruginosa* as compared to coumermycin-treated *P. aeruginosa*.

This case study underscores an important lesson in understanding how antibiotics, even when binding to the same target in the pathogen, but belonging to distinct structural classes, can have varying effects on pathogenic processes. This case study also informs our group with respect to ongoing research projects in the lab. Our team has previously demonstrated that when combined with antibiotics, the microRNA let-7b-5p exhibits additive effects in suppressing biofilm formation in *P. aeruginosa* (59). The findings from this analysis, facilitated by E.PathDash, guide our group in conducting preliminary studies on *P. aeruginosa* to assess the efficacy of various antibiotics. Identifying the most potent biofilm-inhibiting antibiotic can maximize the additive effect of a therapeutic let-7b-antibiotic combo.

## DISCUSSION

E.PathDash facilitates re-analysis of gene expression data from pathogens clinically relevant to respiratory diseases, including a total of 48 studies, 548 samples, and 404 unique treatment comparisons. E.PathDash has several key features that make it a valuable resource for hypothesis generation that also enhances data reusability and

generates reproducible analyses. First, E.PathDash provides multiple entry points into the data sets within its compendium; the user can drill down into a single data set related to a bacterial species of interest, search for activated pathways across data sets, or compare pathway expression between data sets and treatments. The case studies highlighted in this paper demonstrate how this allows researchers to follow an investigative path through multiple questions, all within a single interface, and identify research data sets that show activation of a pathway of interest.

Second, E.PathDash makes the cleaned raw data and analysis results available for download. This promotes transcriptomic analyses beyond the capabilities built into the application and cuts down on the time investigators need to spend preparing publicly available data for reuse (from many hours to seconds).

Third, the application links gene-level and pathway-level analyses, which allows researchers to quickly extend the conclusions of pathway activation to expression data at the individual gene level. This can be done by rendering a volcano plot of genes in a pathway using the "KEGG Pathway Explorer" or "GO Term Explorer" page, as described in Results. In both case studies, E.PathDash was used to identify how genes within a significantly activated pathway were expressed, which led to mechanistic hypotheses regarding pathway activation. Therefore, insights facilitated by E.PathDash can help generate ideas for future experiments.

E.PathDash was developed as a tool that expands upon ESKAPE Act PLUS and CF-Seq, other applications developed by our group that facilitate re-analysis of omics data. This relationship promotes the use of all three applications together, expanding on the hypotheses one could generate from any individual application. ESKAPE Act PLUS performs pathway analysis on user-supplied differential gene expression data sets, using the same binomial test and pathways in E.PathDash. The use of a consistent methodology in both applications allows users to connect transcriptional patterns in their own data sets to those identified by E.PathDash in its compendium of publications. Additionally, CF-Seq and E.PathDash use the same differential gene expression analysis pipeline and contain some of the same publicly available data sets. Therefore, users can explore differential gene expression patterns for those data sets also included in CF-Seq beyond what is communicated by the volcano plots in E.PathDash.

Given the application's relationship to ESKAPE Act PLUS, the data sets were restricted to pathogens included in the analytical pipeline for ESKAPE Act PLUS. This decision restricted the number of RNA-Seq data sets included in the application. An area of future work is to increase the number of data sets and compatible pathogens, which could be done by expanding the number of species and strains for which ESKAPE Act PLUS has pathway data.

In its implementation of pathway activation analysis, ESKAPE Act PLUS assumes that under the null hypothesis the split of up and down regulated genes is 50% (36). This assumption ignores inherent differences between expression levels across RNA-Seq data sets and could be made more sophisticated by using the overall rate of induction within the data set as the null hypothesis ratio. Subsequently, the magnitude of deviation between the overall induction rate and the induction rate among genes within a pathway could determine significant pathway activation or repression.

It is important to acknowledge that E.PathDash and other methods to interrogate archived data are hypothesis generating and should not be used to draw absolute conclusions about responses to experimental conditions. Additionally, the distillation of biological processes into large networks of genes represented by KEGG pathways and GO terms obscures interactions between such processes and may preclude the identification of significantly activated smaller, organism-specific regulatory networks (66, 67). Despite these caveats our group has published several studies where we mined publicly available data and performed pathway analysis, through which new hypotheses were identified and confirmed by laboratory experiments (3, 68–71).

Users of E.PathDash should also note that data sets in the compendium were not generated from experiments that shared wet bench or computational protocols in their

data collection pipeline. Thus, any direct comparisons between data sets should consider the underlying study design differences that could have contributed to the analysis results.

## MATERIALS AND METHODS

### Data collection and cleaning

Figure 8 shows an overview of the data collection, cleaning, and analysis pipelines. Data collection for E.PathDash, represented by the orange arrows in Fig. 8, happened in two different stages. First, data sets were extracted from the compendium originally compiled for the CF-Seq application, which contained relevant data sets published on GEO through July 2021 . Second, the GEO was queried for data sets published between August 2021 and December 2023.

The first wave of data collection started with the compendium of RNA-Seq data sets from the GEO that were originally compiled for the development of CF-Seq. The CF-Seq data collection process restricted data sets to clinically relevant CF pathogens identified by a literature review (38). Additional requirements included RNA-Seq files need to be tables of raw gene counts in formats compatible with R's file functions and the differential expression analysis package edgeR, and study metadata has to define sample groups

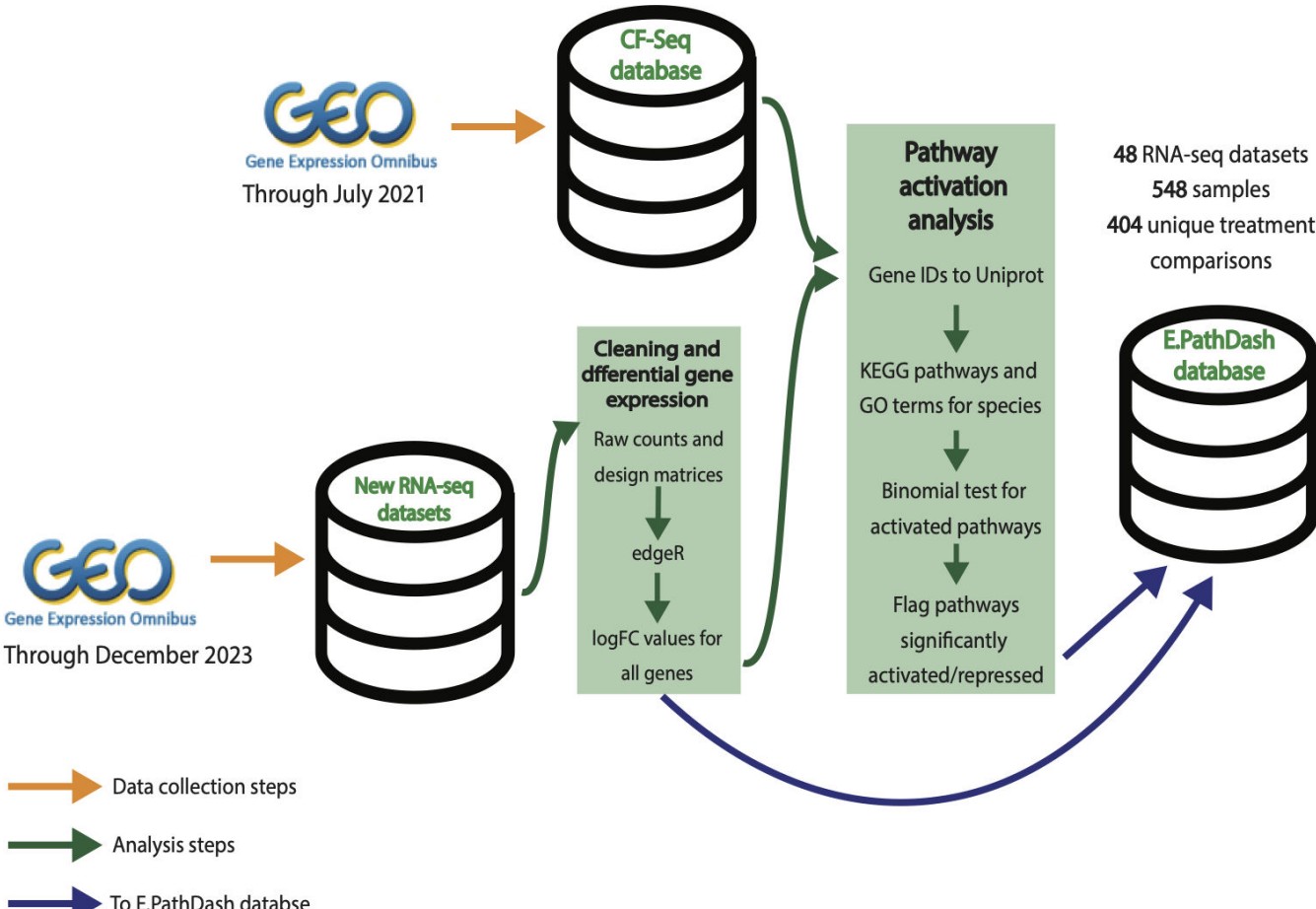

**FIG 8** Data collection, cleaning, and analysis pipeline for the creation of the E.PathDash application database. All RNA-Seq data sets originated from GEO. Data uploaded after July 2021 (not included in the CF-Seq application database) were run through the CF-Seq differential expression analysis pipeline. Subsequently, all data sets were run through the ESKAPE pathway activation analysis pipeline, and both differential analysis and pathway activation results were pulled into the E.PathDash application database.

for the purpose of differential gene expression analysis. These requirements allowed automated processing of the data sets.

Raw count tables and differential gene expression analysis results, obtained using the CF-Seq analysis pipeline, were extracted from the CF-Seq compendium for use in E.PathDash. The extraction process was conducted with R scripts run on R version 4.2.1 (72). The scripts filtered the data sets to those pathogens compatible with the ESKAPE Act PLUS application (36), which was leveraged to conduct pathway activation analysis. Those pathogens were *Pseudomonas aeruginosa* (strains PA14 and PAO1), *Bacteroides thetaiotaomicron* (strain VPI-5482), *Staphylococcus aureus* (strains Newman and USA300), and *Streptococcus sanguinis* (strain SK36). These pathogens have been shown to play an important role in a variety of respiratory diseases (6–15) and can be affected in the airway by pertinent environmental exposures like air pollution (73).

The second wave of data collection pulled data sets published on the GEO from August 2021 through December 2023 that met the same organism, file format, content, and sample group requirements outlined above. The scripts used to do this can be found in the Git repository associated with this publication (https://github.com/DartCF/EPathDash). Specifically, the scripts used the Entrez programming utilities (74) from the NCBI to query the GEO for data sets where the organism was one of the four pathogens of interest, the type was "expression profiling by high throughput sequencing" (identifies RNA-Seq data sets), and the release date was between 1 August 2021 and 31 December 2023. From the collection returned by the Entrez queries, we manually identified data sets that had raw count data in the formats compatible with R's file functions and sample group definitions. Differential gene expression analysis for the new data sets was conducted using the CF-Seq analysis scripts. The final compendium, including data sets from both collection waves, consisted of 48 studies, 548 samples, and 404 unique treatment comparisons across the four different bacterial species.

For the purpose of differential gene expression and pathway analyses, each data set in our final compendium needed a corresponding design matrix that mapped samples to experimental conditions defined in the study. Each design matrix was created manually using sample information from the GEO, ensuring a consistent format. This effort and the development of data set retrieval scripts required significant time upfront that we believe returns time to the researcher using E.PathDash.

ESKAPE Act PLUS, used to conduct pathway activation analysis, uses the KEGGREST R package (75) from Bioconductor to map genes to KEGG pathways for the purpose of the analysis (36). The KEGGREST mappings use UniProtKB (UniProt Knowledgebase) gene encodings, requiring all gene identifiers to be translated to UniProtKB . Metadata for each study was reviewed to identify the gene encoding schema, and the UniProt Consortium's web-based gene ID mapping tool was used to translate gene IDs to UniProtKB (76). Gene IDs in their native encoding were extracted from the RNA-Seq data sets and written to CSV files, which were uploaded to the UniProt ID mapping tool. Batches for each encoding schema were run separately because the translation tool does not dynamically detect the gene encoding schema and requires the input schema to be manually specified. Results consisted of the original gene identifier and the corresponding UniProt ID. This dictionary was used to translate gene IDs for pathway activation analysis. The scripts used to conduct the data pre-processing can be found in the Git repository cited previously.

Within the 48 studies included in the final compendium, 31 of the studies had multiple rows with the same UniProt identifier but different logFC values for differential expression. This inconsistency was a result of there being rows in the raw count data for an ordered locus name and gene name corresponding to the same protein. To resolve this, we chose to keep only the information for the identifier associated with the smaller *P*-value for differential gene expression, optimizing the sensitivity of the downstream pathway activation analysis. In general *P*-values for these unique rows were similar.

After data collection and cleaning, the data structure used for pathway activation analysis consisted of differential gene expression results for each unique treatment

comparison within each data set. Each row of the dataframe consisted of UniProt gene identifier, logFC value, and adjusted *P*-value (corrected for multiple hypotheses using the FDR method).

## Data analysis

Differential gene expression analysis for all data sets was conducted using the R package edgeR (77, 78). After removing low-expression genes and normalizing library sizes, count matrices were fit to a log-linear model using glmQLFit, and gene-wise tests for differential expression were conducted using glmQLFTest. The model and the corresponding statistical test require there be multiple replicates of each experimental condition. Therefore, downstream comparisons made by E.PathDash represent transcriptional differences for groups of samples in the specified treatment conditions. To see the full implementation of the edgeR pipeline that was used for the RNA-Seq data sets in this compendium, refer to the data setup script in the Git repository (https://github.com/DartCF/cf-seq).

Pathway activation analysis was conducted using the same pipeline implemented in ESKAPE Act PLUS (36). We use a binomial test to identify significantly activated or repressed pathways between treatment groups. The binomial test uses only the distribution of positive and negative logFC values for genes in a pathway, thus preventing any single gene from driving the statistical analysis (36). The test's null hypothesis assumes that an equal proportion of genes will be activated (positive logFC value) and repressed (negative logFC value) under random conditions. E.PathDash uses the binom.test function in R's stats package (72) to determine if the observed proportion of genes with positive logFC values in a given pathway differs significantly from 50%. If this difference has a FDR-corrected *P*-value < 0.05, the pathway is labeled as significantly activated or repressed.

An advantage of this method of pathway analysis is that it accounts for all genes within a pathway (and present in the given RNA-Seq data set), not just those that meet a threshold of differential expression between treatment groups. Therefore, information from genes on the edges of a threshold for statistical significance or logFC value is not lost. Another common method to identify activated pathways, over-representation analysis (ORA) for pathway enrichment, restricts the analyzed genes to those with statistically significant differential expression between experimental groups (79).

Pathway activation analysis was conducted for KEGG pathways and GO terms. KEGG and GO are databases annotated by domain experts that map genes to functional groups (41–44). Pathway activation analysis results for each treatment comparison within each data set was stored in the final database for E.PathDash. The stored results include (i) KEGG or GO path name, (ii) raw and FDR-corrected *P*-values, (iii) binomial test statistic, and (iv) median logFC value of genes in the pathway. The script to run pathway activation analysis can be found in the Git repository specified previously.

In addition to differential gene expression and pathway activation analyses, E.PathDash has data that link each KEGG pathway and GO term to their constituent genes, which was retrieved using the EnrichmentBrowser package (80). Using this information, the E.PathDash interface links pathway activation analysis and differential gene expression analysis results.

The data collection, cleaning, and analysis were performed offline in order to minimize the amount of processing the application needed to do in real time, facilitating a better user experience. The collection and analysis steps were split into the two R scripts mentioned previously, which were run in succession on the compiled data sets. The scripts utilize several R packages in addition to the KEGGREST and EnrichmentBrowser packages: "stringr" for string manipulation, "readxl" for file ingestion, and "tidyverse" for data frame manipulation (81–83).

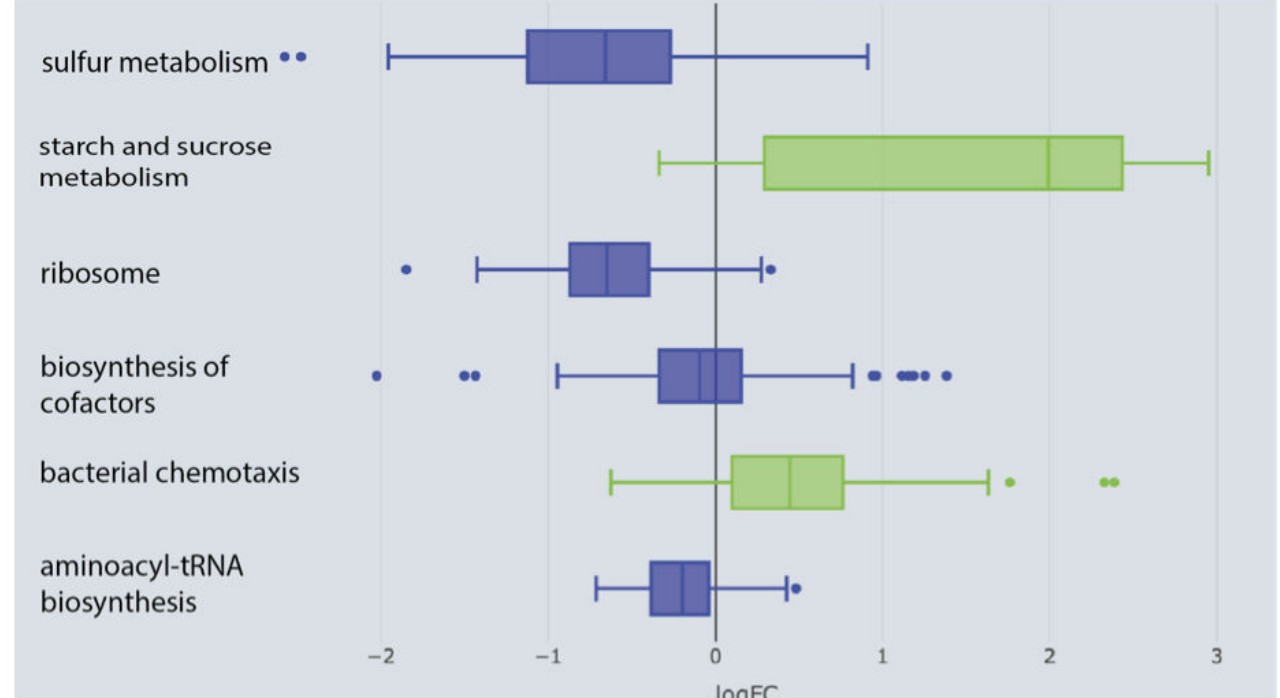

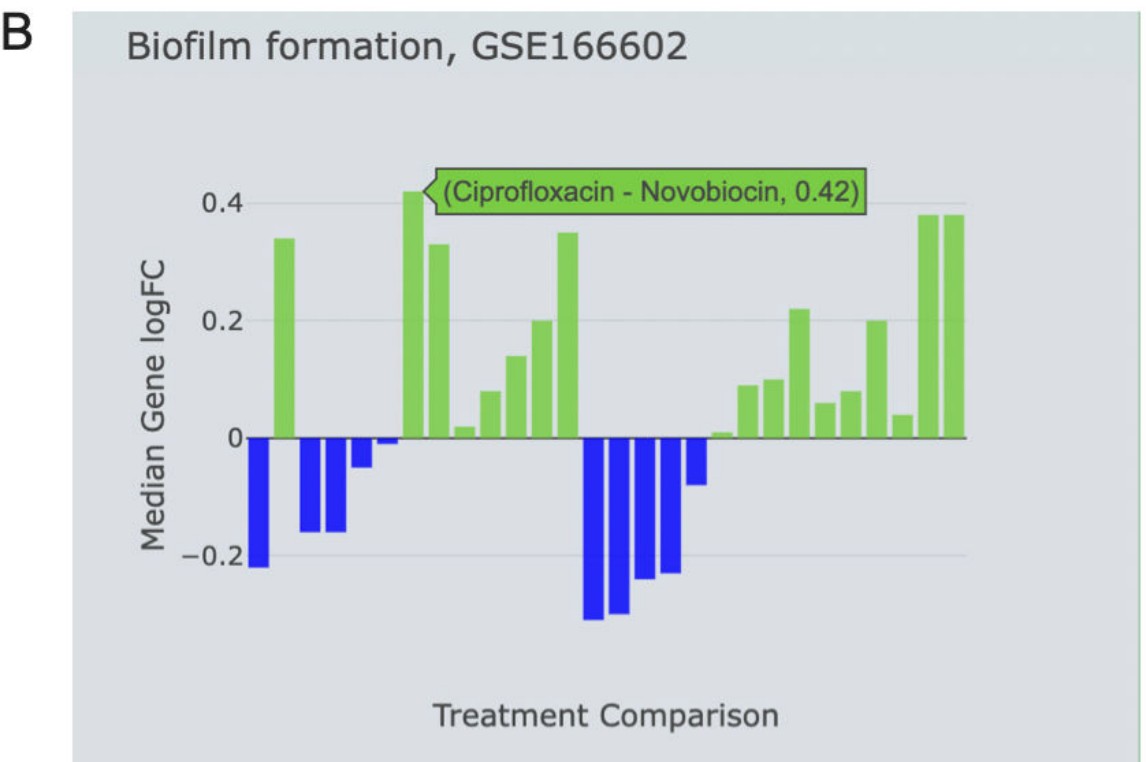

**FIG 9** (A) Boxplot showing pathway analysis results for GEO data set GSE142448, which measures responses of *P. aeruginosa* to manuka honey and its components. The plot shows distributions of the gene logFC values for KEGG pathways differentially expressed between artificial honey and manuka honey, which has a well-known antibacterial effect. (B) Bar plot showing how the KEGG biofilm formation pathway is expressed across all treatment comparisons in GEO data set GSE166602, which exposed *P. aeruginosa* to different gyrase inhibitors.

## Application design

All the application code for E.PathDash is contained in the app.R file, which has two components, "ui" and "server," that define a reactive web application. The code in the "ui" component is responsible for the implementation of the interface. In E.PathDash, this consists of a menu sidebar, a header, and four main dashboards (Fig. 2). The interface is rendered as a single HTML page and elements are hidden or shown based on the application state. The interface components are implemented by the "shinydashboard" R package (84).

The "server" component loads the backend database and defines the results that are returned to the UI in response to how a user interacts with the application. The server code is split into sections that correspond to the four dashboards. Together with the "ui" component, this code utilizes several R libraries for plotting and design: "plotly" for all application plots and "shinycssloaders," "shinyBS," and "shinyjs" for design functionality (85–88). For additional design customization, the app.R file loads a custom CSS design file.

## Pathway visualizations

In addition to displaying pathway analysis results in a table format, E.PathDash shows visualizations that enhance the user's ability to compare different pathway activation or repression patterns within and across studies.

Boxplots show KEGG pathway and GO term activation and repression within a single study (Fig. 9A). The boxplots capture the distribution of the logFC values for genes within each statistically significant activated/repressed KEGG pathway and GO term (adjusted $P$-value < 0.05). Showing the distribution of logFC values for genes within the pathway communicates repression/activation strength between treatment groups. The distributions are colored based on whether the pathway is activated or repressed (positive or negative median logFC value).

When comparing pathway activation between studies, a bar chart shows pathway activation patterns across all possible treatment comparisons within each study (Fig. 9B). Plots show the median logFC value for the genes within the KEGG pathway or GO term, and the treatment comparison and median logFC value are displayed in a popup message upon hovering over each bar in the chart.

## ACKNOWLEDGMENTS

This work was supported by the Cystic Fibrosis Foundation (STANTO19G0, STANTO20PO, STANTO19R0), the National Institutes of Health (P30-DK117469, R01HL151385, P20-GM113132), and the Flatley Foundation.

L.T. wrote the publication except for the user stories and developed the E.PathDash application. L.T., S.L.N., and T.H.H. conceived of the application, and T.H.H. provided guidance throughout the development and publication drafting processes. S.S., G.D., and C.E.F. tested the application and provided feedback on its utility, and S.S. and G.D. wrote user stories for this publication. B.A.S. contributed valuable feedback during publication drafting and provided primary funding for this project. All authors reviewed drafts of the publication.

## AUTHOR AFFILIATIONS

[1]Department of Microbiology and Immunology, Geisel School of Medicine, Dartmouth College, Hanover, New Hampshire, USA
[2]The Jackson Laboratory for Genomic Medicine, Farmington, Connecticut, USA
[3]Lewis-Sigler Institute for Integrative Genomics, Princeton University, Princeton, New Jersey, USA

## AUTHOR ORCIDs

Lily Taub http://orcid.org/0009-0008-6625-6216
Thomas H. Hampton http://orcid.org/0000-0003-0543-402X
Sharanya Sarkar http://orcid.org/0000-0002-2459-8620
Georgia Doing http://orcid.org/0000-0002-0835-6955
Samuel L. Neff http://orcid.org/0000-0002-5993-8445
Carson E. Finger http://orcid.org/0000-0002-0335-4547
Kiyoshi Ferreira Fukutani http://orcid.org/0000-0003-2223-0918
Bruce A. Stanton http://orcid.org/0000-0002-1661-407X

## FUNDING

| Funder | Grant(s) | Author(s) |
|---|---|---|
| Cystic Fibrosis Foundation (CFF) | STANTO19G0,STANTO20PO,STANTO19R0 | Lily Taub |
| | | Thomas H. Hampton |
| | | Sharanya Sarkar |
| | | Kiyoshi Ferreira Fukutani |
| | | Bruce A. Stanton |
| HHS \| National Institutes of Health (NIH) | P30-DK117469,R01HL151385,P20-GM113132 | Lily Taub |
| | | Thomas H. Hampton |
| | | Sharanya Sarkar |
| | | Kiyoshi Ferreira Fukutani |
| | | Bruce A. Stanton |
| Flatley Foundation | | Thomas H. Hampton |
| | | Sharanya Sarkar |
| | | Kiyoshi Ferreira Fukutani |
| | | Bruce A. Stanton |

## AUTHOR CONTRIBUTIONS

Lily Taub, Conceptualization, Data curation, Software, Writing – original draft | Thomas H. Hampton, Conceptualization, Writing – original draft, Writing – review and editing | Sharanya Sarkar, Validation, Writing – original draft, Writing – review and editing | Georgia Doing, Validation, Writing – original draft, Writing – review and editing | Samuel L. Neff, Conceptualization, Writing – review and editing | Carson E. Finger, Validation, Writing – review and editing | Kiyoshi Ferreira Fukutani, Writing – review and editing | Bruce A. Stanton, Funding acquisition, Writing – review and editing

## DATA AVAILABILITY

Raw count tables, constructed design matrices, and additional study metadata files for all GEO data sets included in this version of E.PathDash are available in the Git repository at https://github.com/DartCF/EPathDash/tree/main/GEO_Datasets. All application code and data processing and analysis scripts are available in the Git repository at https://github.com/DartCF/EPathDash. The application is hosted in a cloud computing environment maintained by Dartmouth Research Computing Infrastructure and can be accessed at scangeo.dartmouth.edu/EPathDash. In its current version, E.PathDash utilizes the following R packages: KEGGREST (v 1.32.0), EnrichmentBrowser (v 2.28.0), stringr (v 1.5.0), readxl (v 1.4.3), tidyverse (v 1.3.2), shinydashboard (v 0.7.2), plotly (v 4.10.1), shinycssloaders (v 1.0.0), shinyBS (v 0.61.1), and shinyjs (v 2.1.0). The application runs on R version 4.3.3.

## ADDITIONAL FILES

The following material is available online.

### Supplemental Material

**Supplemental material (mSystems01030-24-s0001.pdf).** Figures S1-S7 and Tables S1, S3, and S4.
**Table S2 (mSystems01030-24-s0002.xlsx).** Differential gene expression results for data set GSE148597.

### Open Peer Review

**PEER REVIEW HISTORY (review-history.pdf).** An accounting of the reviewer comments and feedback.

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
