## [Reviewer comments · mSystems]

E.PathDash, pathway activation analysis of publicly available pathogen gene expression data

Lily Taub, Thomas Hampton, Sharanya Sarkar, Georgia Doing, Samuel Neff, Carson Finger, Kiyoshi Ferreira Fukutani, and Bruce Stanton

Corresponding Author(s): Lily Taub, Dartmouth College Geisel School of Medicine

Review Timeline:

Submission Date:

August 1, 2024

Accepted:

September 20, 2024

Editor: David Rasko

Reviewer(s): The reviewers have opted to remain anonymous.

Transaction Report:

DOI: <https://doi.org/10.1128/msystems.01030-24>

Re: mSystems01030-24 (E.PathDash, pathway activation analysis of publicly available pathogen gene expression data)

Dear Dr. Lily Taub:

The revised manuscript is acceptable for publication.

Your manuscript has been accepted, and I am forwarding it to the ASM production staff for publication. Your paper will first be checked to make sure all elements meet the technical requirements. ASM staff will contact you if anything needs to be revised before copyediting and production can begin. Otherwise, you will be notified when your proofs are ready to be viewed.

Sincerely,
David Rasko
Editor
mSystems

Reviewer #2 (Comments for the Author):

The authors have satisfactorily addressed the comments and made sufficient changes to the manuscript. The manuscript is adequate.